# Remarkable Remission Rate and Long-Term Efficacy of Upfront Metronomic Chemotherapy in Elderly and Frail Patients, with Diffuse Large B-Cell Lymphoma

**DOI:** 10.3390/jcm11237162

**Published:** 2022-12-01

**Authors:** Guido Bocci, Sabrina Pelliccia, Paola Orlandi, Matteo Caridi, Marta Banchi, Gerardo Musuraca, Arianna Di Napoli, Maria Paola Bianchi, Caterina Patti, Paola Anticoli-Borza, Roberta Battistini, Ivana Casaroli, Tiziana Lanzolla, Agostino Tafuri, Maria Christina Cox

**Affiliations:** 1Department of Clinical and Experimental Medicine, School of Medicine, University of Pisa, 56126 Pisa, Italy; 2UOC Ematologia, Azienda Ospedaliera Universitaria Sant’Andrea, 00189 Rome, Italy; 3Division of Hematology and Clinical Immunology, Department of Medicine, University of Perugia, 06125 Perugia, Italy; 4Hematology Unit, Istituto Scientifico Romagnolo per lo Studio e la Cura dei Tumori (IRST) Srl—IRCCS, 47014 Meldola, Italy; 5UOC Anatomia Patologica, Azienda Ospedaliera Universitaria Sant’Andrea & Department of Clinical and Molecular Medicine Sapienza University, 00185 Rome, Italy; 6UOC Oncoematologia, Azienda Villa Sofia-Cervello, 90146 Palermo, Italy; 7UOC Ematologia, Azienda Ospedaliera San Giovanni-Addolorata, 00184 Rome, Italy; 8UOC Ematologia, Azienda Ospedaliera San Camillo, 00152 Rome, Italy; 9Haematology Department, San Gerardo Hospital Monza, 20900 Monza, Italy; 10UOC Medicina Nucleare, Azienda Ospedaliera Universitaria Sant’Andrea, 00189 Rome, Italy; 11Hematology Unit, Fondazione Policlinico Tor Vergata, 00133 Rome, Italy

**Keywords:** metronomic chemotherapy, chemo-free, DLBCL, diffuse-large-b-cell-lymphoma, elderly, frail, comprehensive geriatric assessment

## Abstract

The upfront treatment of very elderly and frail patients with diffuse large B-cell lymphoma (DLBCL) is still a matter of debate. Herein, we report results of the metronomic all-oral DEVEC [prednisolone/deltacortene^®^, vinorelbine (VNR), etoposide (ETO), cyclophosphamide] combined with i.v. rituximab (R). This schedule was administered as a first line therapy in 22 elderly/frail DLBCL subjects (median age = 84.5 years). In 17/22 (77%) patients, the Elderly-IPI-score was high. After a median follow-up of 24 months, 15 patients had died: seven (50%) for causes unrelated to DLBCL or its treatment, six (40%) for progression, and two (13%) for multiorgan failure. Six treatment-pertinent serious-adverse-events occurred. At the end of induction, 14/22 (64%) achieved complete remission; overall survival and event-free survival at 24 months were both 54% (95% CI = 32–72%), while the time to progression was 74% (95% CI = 48–88%). Furthermore, antiproliferative and proapoptotic assays were performed on DLBCL/OCI-LY3 cell-line using metronomic VNR and ETO and their combination. Both metronomic VNR and ETO had concentration-dependent antiproliferative (IC50 = 0.036 ± 0.01 nM and 7.9 ± 3.6 nM, respectively), and proapoptotic activities in DLBCL cells. Co-administration of the two drugs showed a strong synergism (combination index < 1 and dose reduction index > 1) against cell proliferation and survival. This low-dose schedule seems to compare favourably with intravenous-CHEMO protocols used in the same subset. Indeed, the high synergism shown by metronomic VRN+ETO in in vitro studies, explains the remarkable clinical responses and it allows significant dose reductions.

## 1. Introduction

Diffuse large B-cell lymphoma (DLBCL) is an aggressive and most common type of lymphoma, with an incidence of about 7 cases per 100.000 persons per year [1]. As up to 70% of patients, who are <80 years and fit, can be cured by the combination of Rituximab-CHOP [2] and the use of intravenous chemotherapy (iv-CHEMO), which remains a pillar of upfront treatment. However, in patients ≥80 years and in frail subjects, alternative approaches are urgently needed [3]. In fact, vulnerable elderly, currently represents a fast-growing subset of DLBCL [4] and their management, is a matter of concern in clinical practice. As none of the iv-CHEMO schedules so far experimented in this subset [5,6,7] have become the standard of care.

Conversely, ongoing clinical trials based on immunomodulators [8], new monoclonal antibodies [9] and small molecules targeting pathogenetic pathways [10] will hopefully help a shift towards therapeutic schedules defined as chemo-free, in vulnerable DLBCL. As a matter of fact, results from a few phase II trials, based on chemo-free combinations, have already reported encouraging data [11,12].

Worthy of note, in the changing landscape of DLBCL treatment, the potential of metronomic chemotherapy (mCHEMO), which is a different way of delivering and utilising well-known cytotoxic drugs, has been very little explored [13]. This lack of knowledge, compared with solid tumours [14], is presumably due to the high success rate of standard iv-CHEMO schedules in fit patients.

In 2011 we started to experiment in vulnerable non-Hodgkin lymphoma (NHL) patients, the DEVEC [Deltacortene^®^/Prednisolone (PDN), Etoposide (ETO), Vinorelbine (VNR), Cyclophosphamide (CTX)] schedule, which is an all-oral metronomic poli-chemotherapy combination. Herein, we report the updated results of this metronomic schedule used upfront in elderly/frail DLBCL patients. We also support the clinical results with data from in-vitro experiments, which assessed the activity of metronomic VNR and metronomic ETO combinations in DLBCL cells.

## 2. Methods and Patients

### 2.1. In Vitro Experiments

#### Cells, Drugs and Reagents

OCI-LY3, a DLBCL cell line, was purchased from DMSZ (Leibniz Institute DSMZ-German Collection of Microorganisms and Cell Cultures GmbH, Braunschweig, Germany; ACC 761) and grown in an 80% RPMI-1640 medium supplemented with 20% heat-inactivated FBS, antibiotics, and L-glutamine 2 mM. The cells grow singly or in clumps in suspension and were maintained in tissue culture flasks and kept in a humidified atmosphere of 5% CO_2_ at 37 °C.

VNR and ETO were obtained from Selleckchem (DBA Italia, Milan, Italy) and were diluted from a 10 mM stock solution (in 100% dimethylsulfoxide) for in vitro studies. Vehicle-treated controls received the same concentration of dimethylsulfoxide in the media as cells of the highest concentration of VNR and ETO.

Sterile plastics for cell culture were from Costar (Cambridge, MA, USA), whereas L-glutamine, antibiotics, RPMI-1640 medium, foetal bovine serum (FBS) and all the other cited reagents were purchased from Sigma Aldrich SRL (Milan, Italy).

### 2.2. Antiproliferative Assay of Metronomic Vinorelbine and Etoposide

The assay was performed as previously described [15] with minor modifications. OCI-LY3 cells were exposed for 144 h with metronomic VNR (0.001–10 nM) and ETO (0.025–100 nM), or with vehicle alone. VNR or ETO were added every 48 h or 24 h, respectively, to mimic the clinical schedules. At the end of the treatment, the viable cells, assessed by trypan blue dye exclusion, were counted with a hemocytometer. The drug concentrations that inhibited cell proliferation by 50% (IC_50_) versus vehicle-treated cells were determined by a nonlinear regression fit of the mean values obtained in triplicate experiments (at least 9 wells for each concentration).

### 2.3. Determination of Synergism between Metronomic Vinorelbine plus Etoposide on OCI-LY3 Cells

The determination of synergism between metronomic vinorelbine and metronomic etoposide on OCI-LY3 cells in vitro was performed to report the type of pharmacodynamic interaction between the drugs and the possible dose reduction in each drug when used in this combination schedule.

The concomitant combination of metronomic VNR and ETO was investigated at different concentrations with a fixed molar ratio of 1:100, respectively. Synergism was calculated by the combination index (CI) and the dose reduction index (DRI) method [16], where CI < 1, CI = 1, and CI > 1 indicate synergism, additive effect, and antagonism, respectively. The DRI represents the theoretical magnitude of concentration decrease that could be obtained for each drug in combination to obtain the same effect of each drug alone. Briefly, synergism, additivity, or antagonism for metronomic vinorelbine plus metronomic etoposide was calculated on the basis of the multiple drug-effect equations and quantitated by the combination index (CI). Based on the classic isobologram, the CI value was calculated as:CI = [(D)_1_/(D_x_)_1_] + [(D)_2_/(D_x_)_2_]

As an example, at the 90% inhibition level, (D_x_)_1_ and (D_x_)_2_ are the concentrations of metronomic vinorelbine and metronomic etoposide, respectively, that induce a 90% inhibition of cell growth; (D)_1_ and (D)_2_ are the concentrations of metronomic vinorelbine and metronomic etoposide in combination that also inhibits cell growth by 90% (isoeffective as compared with the single drugs alone). The dose-reduction index (DRI) defines the degree of dose reduction that is possible in combination with a given degree of effect as compared with the concentration of each drug alone, and it was calculated as follows:(DRI)_1_ = (D_x_)_1_/(D)_1_ and (DRI)_2_ = (D_x_)_2_/(D)_2_

The CI and DRI indexes were estimated by the CalcuSyn v.2.0 software (Biosoft, Cambridge, UK). Furthermore, the synergistic, additive, and antagonistic effect of the drug combination was also mapped with the Loewe additivity model, using the Combenefit software (v.2.021) [15].

### 2.4. Apoptosis Assay

To quantify the extent of apoptosis, 3 × 10^5^ OCI-LY3 cells were treated for 48 h with metronomic VNR, metronomic ETO, and their concomitant combination at the experimental IC_50_ or with vehicle alone as a control. At the end of the experiment, control and treated cells were analysed using the Cell Death Detection ELISA Plus Kit (Roche, Basel, Switzerland) as per the manufacturer’s instructions. The optical density was measured using a Multiskan Spectrum microplate reader (Thermo Labsystems, Milan, Italy) set to a wavelength of 405 nm (with a wavelength of 490 nm correction). All the absorbance values were plotted as a percentage of apoptosis relative to vehicle-treated cells which are labelled as 100%. All experiments were repeated three times with at least three replicates per sample.

### 2.5. Statistical Analysis of In Vitro Studies

The investigators responsible for data analysis were blinded to which samples represented treatments and controls. The results (mean ± SEM) of all the experiments were analysed with ANOVA, followed by the Student-Newman-Keuls test. The statistical significance was set at *p* < 0.05. Statistical analyses were carried out by the GraphPad Prism software package version 5.0 (GraphPad Software, Inc., San Diego, CA, USA).

### 2.6. Clinical Study

This is a multicentre, observational study involving six clinical centres, which adopted the all-oral DEVEC mCHEMO schedule to treat vulnerable elderly NHL patients. From 25 October 2015 to 31 December 2017, data were retrospectively collected, while from 1 January 2018 to 30 September 2022 they were prospectively recorded. The schedule was offered to patients who were considered unfit for treatment with curative intent as an alternative to IV-CHEMO. The first patient described in this study started treatment on 16 October 2015, while the last follow-up was carried out on 30 September 2022. Rituximab (R) was added to DEVEC in B-cell NHL expressing the CD20-antigen. Only subjects with a confirmed diagnosis of DLBCL, who were treated up-front with the R-DEVEC schedule were considered for this report. This 28-days schedule (Appendix A), was repeated for six cycles, as previously described [17]. The starting dose of ETO was empirically fixed to 14 days, with the intention to taper ETO doses within the first cycle of mCHEMO, and to allow haematological values within the established threshold. The first four Rituximab (R) infusions were administered weekly at a dose of 375 mg/m^2^, starting from day +8 of cycle 1, while the last two doses (5th and 6th) were both given at an interval of 21–28 days. There was no need of pre-phase with 50 mg PDN [18]. The Hans’ algorithm [19] was used to classify DLBCL cases. In addition, when immunohistochemistry showed double expression of the genes: MYC (i.e., >40%) and BCL2 (i.e., >50%) or BCL6 (i.e., >40%), FISH analysis was performed to assess if these genes were split. Patients, who achieved at least a partial remission (PR), at the end of induction were offered to continue treatment with additional six maintenance cycles of 21-days and a post maintenance phase (Appendix A). Also, it was explained to patients and their family that maintenance therapy was optional and the benefit of it uncertain. Each cycle was initiated if polymorphonuclear leukocytes (PMN) ≥ 1500, platelet (PLT) ≥ 50,000 and haemoglobin (Hb) ≥ 9.5 gr/dL. Granulocyte colony stimulating factor (G-CSF) and erythropoietin were allowed during the induction cycles. Low molecular weight heparin (LMWH) and low-dose aspirin were administered to patients at high and low-medium risk of thrombosis respectively, during the induction phase. Cotrimoxazole and acyclovir prophylaxis were suggested for all subjects, while ciprofloxacin prophylaxis was started if PMN < 1.0 × 10^9^. Adverse events were coded following the CATCAE v4.03 (https://www.eortc.be/services/doc/ctc/CTCAE_4.03, accessed on 2 July 2022).

Statistical analysis was performed with GraphPad Prism 5 (GraphPad Software, San Diego, CA, USA). Overall (OS), progression-free (PFS) and event-free survival (EFS) analyses were carried out using Kaplan-Meier, and its significance was assessed by the log-rank (Mantel-Cox) test. Treatment stops, before six cycles of induction by any cause, progression, and deaths were all considered events for the EFS analysis. Fisher’s exact test was used to evaluate the impact of significant co-variates on survival.

Caregivers were required to guarantee the proper administration of DEVEC in patients who were very old and frail. Restaging was scheduled by computerized-tomography (CT) scan between the 2nd and 3rd induction-cycles at the end of the induction phase by FDG positron-emission CT-scan (CT-PET) [20] and every six months in the following two years. Last follow-up data were retrieved as of 30 September 2022.

## 3. Results

### 3.1. In Vitro Results

The 144h treatment with metronomic VNR and metronomic ETO showed a significant concentration-dependent inhibitory activity on human OCI-LY3 cell proliferation. The calculated IC_50_s at 144 h was obtained in these cells at 0.036 ± 0.01 nM with VNR, whereas the 50% of cell proliferation was inhibited at a concentration of 7.9 ± 3.6 nM with ETO (Figure 1A). Simultaneous exposure to metronomic VNR and ETO showed synergism (CI > 1) at effect levels exceeding 60% inhibition represented by the fraction of affected cells (Figure 1B). Synergism corresponding to C > 1 always yielded a favourable DRI for both drugs (Table 1). Indeed, in the cases of metronomic VNR or ETO, it could be possible to reduce the concentration of the drug in vitro more than 15-fold or 217-fold, respectively, when the drug is combined to obtain the same 95% level of cytotoxic effects (Table 1). The Loewe analysis of synergism (Figure 1C) confirmed the findings obtained with the Chou method for high percentage of cell proliferation inhibition but also showed a wide area of additivity effect (light green and azure colours) between the 40 and 60% of cell proliferation decrease. Interestingly, synergism and related reductions of drug concentrations are favourable for high level of inhibition of cell proliferation (>90%) to obtain a clinical benefit such as the reduction in the number of tumour cells or the control of neoplastic disease. A representative 95% isobologram of OCI-LY3 cells exposed to concomitant metronomic VNR and ETO for 144 h has been drawn (Appendix A). The IC_95_ values of each drug are plotted on the axes while the line represents the additive effect; the point, reported on the left of the connecting line, reproduces the concentrations of VNR and ETO resulting in 95% growth inhibition of the concomitant combination, indicating synergism.

The proapoptotic activity of the concomitant combination of metronomic VNR and ETO on DLBCL cells was quantified using an ELISA test. The extent of DNA fragmentation was dependent on the exposure of the experimental drug and on the combination. In particular, the Figure 2 shows a significant proapoptotic activity of both single drugs at the IC_50_ already after 48 h of exposure in OCI-LY3 cells, whereas the same drug concentration greatly and significantly enhances the apoptotic signal.

### 3.2. Clinical Results

Twenty-two DLBCL patients started R-DEVEC upfront (Table 2). All subjects were elderly and most were ≥80 years (n = 18/22, 82%). Comprehensive geriatric assessment (CGA), as published by Merli and co-workers [21], was carried out before starting treatment. All patients, but one who was unfit, scored Frail or Super-frail (n = 21/22, 95%) (Table 2). Fifteen patients out of 22 (57%) completed six induction cycles (median = 7 cycles, range 2–18). Only 8 out of 14 (57%) patients who achieved CR, started maintenance cycles (median = 6, range = 4–12). The remaining six patients did not, because of personal or medical choice.

After a median follow-up of 24 months (range 4–80 months), 15 patients had died: seven (47%) for causes unrelated to DLBCL or its treatment (Appendix A), six (40%) for progression, and two (13%) for multiorgan failure. The latter two subjects had stopped treatment following cycles one and two, respectively, because of excessive toxicity. No patient’s death was considered as directly related to treatment toxicity. The first 12 patients started R-DEVEC within 14 days of ETO administration [17]. However, 5/12 (42%) subjects required a reduction in ETO doses during cycles 1 or 2 because of the occurrence of grade ≥3 extra-hematologic toxicities. Therefore, the following 11 subjects started the schedule with only seven days of ETO, while the drug was omitted in all Super-Frail patients (i.e., R-DEVEC-light) from cycle one or two. Also, G-CSF prophylaxis from day 22, was thereafter recommended, in all subjects. Overall, treatment-related serious adverse events (TR-SAE) were recorded in 6/22 subjects (27%; 95% CI = 14–33): three febrile neutropenia, one urosepsis and two cases of pneumonia, respectively. Indeed, only one TR-SAE occurred after the upfront reduction in ETO. Neutropenia and anaemia of grade ≥3 were recorded in 9/22 (41%) and 5/22 (23%), respectively.

At the interim evaluation, the overall response (ORR) and the complete remission rate (CRR) were 77% and 32%, respectively, while at the end of induction the ORR and the CRR were both 64% (Table 3). Overall (OS) (Figure 3A) and event-free (EFS) survivals at 24 months were both 54% 95% CI = 32–72), while time to progression (TTP) was 74% (95% CI = 48–88%) (Figure 3B). Univariate analysis was also carried out to investigate the influence of several parameters on survival (Table 3). International prognostic index (IPI), Elderly-IPI (EPI) [22] and not achieving CR at the end of induction, negatively affected both EFS and OS (Table 3). In addition, there was a negative prognostic trend also for bulky mass ≥7.5 cm, which did not achieve statistical significance (Table 3). Noteworthy, there was no difference in outcome between patients, who following R-DEVEC received and who did not receive maintenance (Table 3). Conversely, the three super-frail subjects, who achieved CR, following R-DEVEC-light, received at least six cycles of maintenance.

## 4. Discussion

This study confirmed our preliminary report regarding metronomic chemotherapy in DLBCL [17]. The main clinical result consists in the ability of upfront R-DEVEC, to achieve long-lasting remission in Elderly/Frail DLBCL. Moreover, based on this extended clinical experience, we were able to provide more accurate, practical suggestions about dosing, duration, and administration of R-DEVEC, in very elderly and frail subjects. More importantly, we report new pharmacological data about this schedule, which rationally supported the clinical results and could help to optimize its use in vulnerable elderly patients.

Indeed, our in vitro data on the human DLBCL cell line OCI-LY3 demonstrated that both metronomic VNR and ETO directly and remarkably inhibited the DLBCL cell proliferation and promoted the apoptotic process. The selection of the OCI-LY3 cell line was based on various characteristics: (i) the origin of the cell was not from a young patient but from a middle-aged man with a stage IV DLBCL that reflects more our population (64% elderly male; 86% stage III-IV); (ii) the cell line is CD20+, CD19+, CD37+ and CD80+ that opens the possibility to future in vitro combination of metronomic chemotherapy with target therapies such as rituximab, tafasitamab or antiCD37 antibodies; (iii) the cell line has been described to harbor 3–4 copies of cMYC and the rearrangements of BCL-2 and BCL-6 [23]. OCI-LY3 cells were extremely sensitive to both VNR and ETO given in vitro metronomically, with calculated IC_50_s in the range of picomolar and nanomolar concentrations, respectively, if maintained constant for a protracted time (i.e., six days). However, more central for clinical translation of the results, their simultaneous metronomic association showed a robust synergistic effect—described with two different methodologies—on OCI-LY3 cells for high percentages of cell proliferation inhibition and exhibited a powerful proapoptotic effect (+142% vs. control). These characteristics suggest that the combination of the two drugs is far more effective than the single treatment. This observation opens to a strong reduction in the doses of both compounds when combined (as witnessed by the DRIs in Table 1) without losing any activity on cancer cells and further reducing the risk of dangerous toxicities. These low and effective concentrations can be easily reached in the plasma of patients administered with metronomic schedules, including DEVEC, as demonstrated by our group and other teams [24,25,26]. Of more interest, Gusella and colleagues reported that metronomic vinorelbine-treated patients who had long-term benefits without toxicity, showed lower VNR concentrations than those who had not [26].

In this series, the higher value of TTP versus OS reflects that a fair percentage of deaths occurred in CR patients; we would highlight that these subjects, due to their old age and burden of comorbidities, often had died for causes unrelated to DLBCL or its therapy. This observation further emphasizes the tolerability and activity of R-DEVEC. Based on our experience, we suggest elderly frail patients receive no more than 7 days of ETO. Alternatively, because of the synergistic activity of ETO and VNR, observed in our in vitro pharmacological experiments, elderly frail may: (1) be treated with a much lower daily dose of ETO for more days; or (2) receive 50 mg of ETO every two days. Conversely, to avoid excessive toxicity, ETO should be omitted in Super-frail subjects, who instead can be safely treated with Rituximab, VNR, CTX and PDN combination (i.e., R-DEVEC-light). Also, G-CSF should be scheduled in all patients from day 21. Worthy of note, peripheral neuropathy, which is an insidious long-term side effect of iv-CHEMO containing vinca-alkaloids, such as R-mini-CHOP, never occurred in patients treated with DEVEC [15,17]. Both high IPI and EPI scores, which were prevalent in this series, had a poor prognostic impact, while the presence of a bulky mass did not achieve statistical significance. Hence, we can speculate that impaired delivery of iv-CHEMO, due to vessels compression, which occurs in tumour mass [27,28,29], may not be an issue with mCHEMO. In fact, the lower and persistent plasma concentrations, allowed by mCHEMO, determine the tumour vascular normalization and the ability to better diffuse in tumour tissues [30]. Interestingly, 3/6 (50%) of Super-frail patients, who were treated with R-DEVEC-light (i.e., devoid of ETO), achieved CR and had a sustained remission following maintenance cycles. Conversely, maintenance, may not be necessary in treatment-naïve patients who achieve CR after six R-DEVEC cycles (i.e., containing ETO). In fact, PFS was not negatively affected by stopping mCHEMO after induction (Table 3). We acknowledge this study has several limitations: (1) not a controlled study; (2) a limited sample size and (3) the lack of information about the overall number of very elderly and frail subjects who started iv CHEMO.

Notwithstanding, our data seems to compare favourably with the seminal work of Peyrade and Co-workers [5]. In fact, following R-miniCHOP the 24-months disease free survival was reported to be slightly less than 60% (vs. 74% for R-DEVEC) [5]. Furthermore, two recent trials based on Rituximab-lenalidomide [11] and Rituximab-lenalidomide-ibrutinib(iR2) [12] combinations, respectively, reported a 12-months PFS of 55%, [11] and a 24-months PFS of 53%, respectively. We acknowledge these comparisons, although intriguing, remain speculative. A prospective, randomized clinical trial is needed to fully legitimize the use of all-oral, non-expensive mCHEMO as a first-line treatment for the vulnerable and elderly. Unfortunately, as no drug of the DEVEC combination is under patent, the making of a prospective trial is currently hampered by a lack of funding.

Indeed, lenalidomide is a cytotoxic compound. Nonetheless, because it is administered orally and at low doses, it is not properly classified as a chemo-free treatment. Hence, lenalidomide, as well as other chemotherapeutic drugs given at metronomic doses, should be more appropriately indicated as chemo-free-like schedules.

Despite low concentrations of cytotoxic drugs, our in vitro data and clinical results rationally supported the use of the metronomic schedule R-DEVEC in elderly, treatment-naïve DLBCL patients. Indeed, this therapeutic approach allows an effective pharmacological activity on cancer cells that translates into a remarkable remission rate and long-term efficacy, with acceptable toxicity and good tolerability.

## Figures and Tables

**Figure 1 jcm-11-07162-f001:**
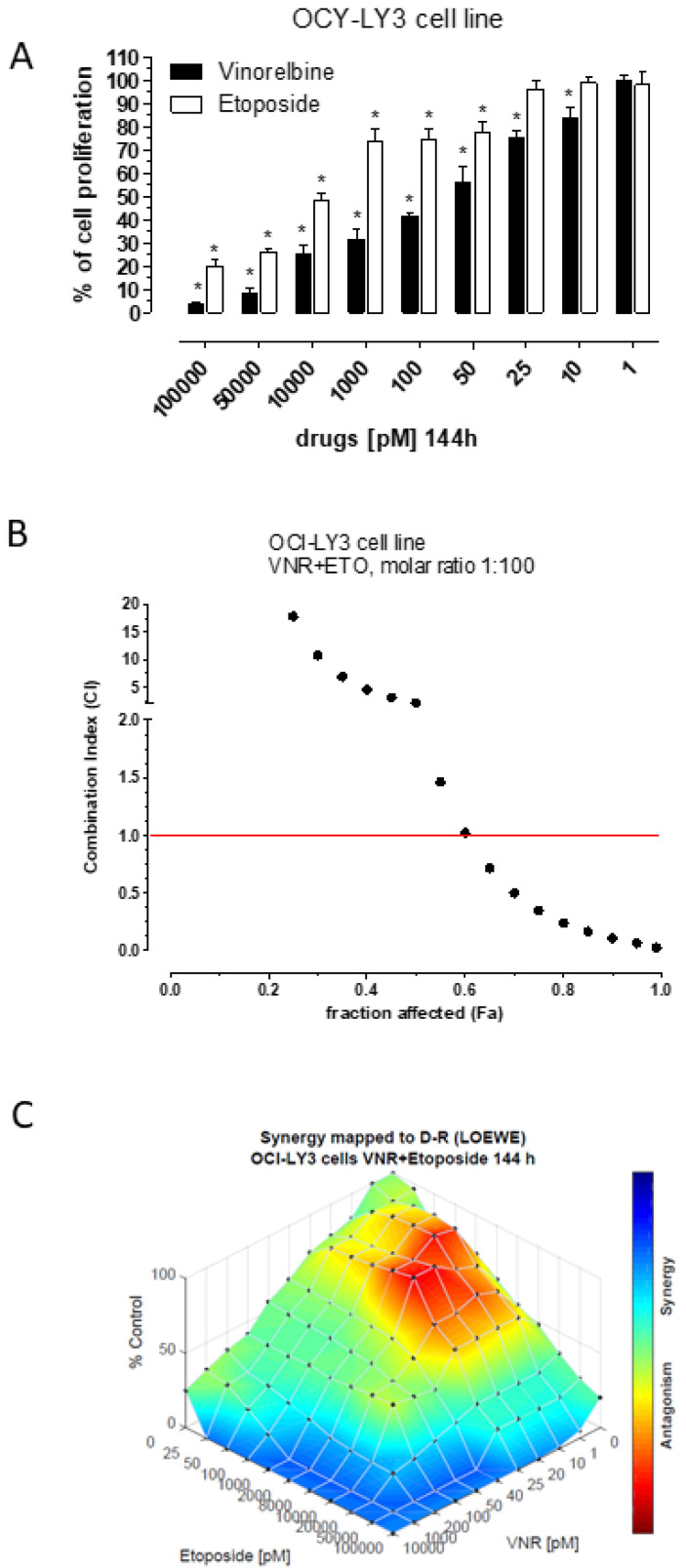
(**A**) Antiproliferative effects of metronomic vinorelbine and metronomic etoposide after 144 h of exposure. Columns and bars, mean values ± SEM, respectively. * *p* < 0.05 vs. vehicle-treated controls. (**B**) Combination index (CI)-fraction affected (Fa) plot of the metronomic vinorelbine (VNR) and etoposide (ETO) combination 144 h at the molar ratio of 1:100 in OCI-LY3 cell line; CI < 1, CI = 1 and CI > 1 indicate synergism, additive effect, and antagonism, respectively. (**C**) The 3-dimensional landscape of the dose matrix of combination responses for metronomic VNR and metronomic ETO based on the Loewe model, where blue reflects evidence of synergy and red represents evidence of antagonism. The model supported synergy of the combination in reducing OCI-LY3 cell line viability. Cell viability was plotted as % control.

**Figure 2 jcm-11-07162-f002:**
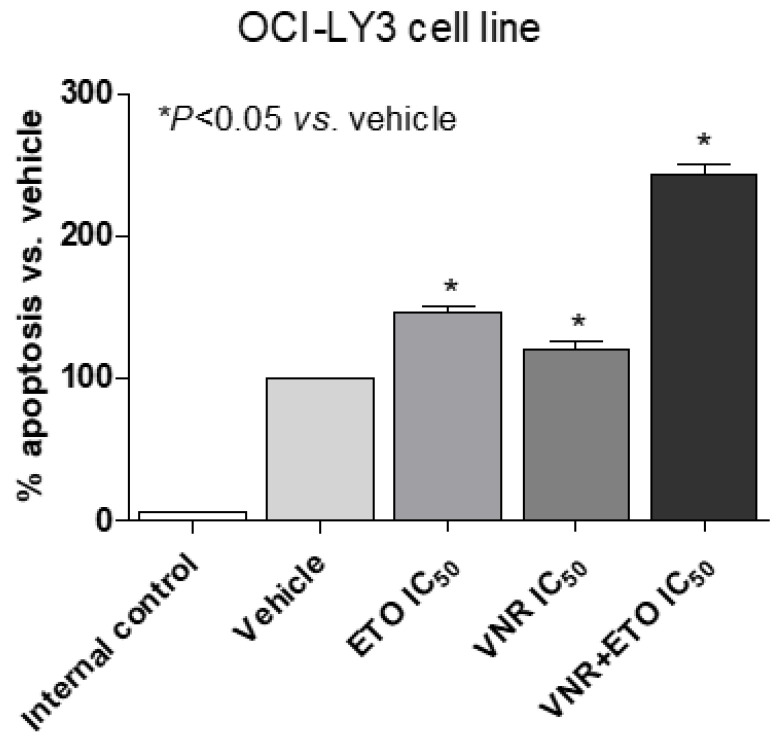
Apoptosis measurements using the Cell Death Detection ELISA Plus kit after 48 h of treatment. Absorbance values are representative of vehicle-treated and metronomic VNR, metronomic ETO and their combination cell cytosolic nucleosomes. All the absorbance values were plotted as a percentage of apoptosis relative to vehicle-treated cells which is labelled as 100%. The internal negative control was provided by the ELISA kit. Columns and bars, mean values ± SD, respectively. * *p* < 0.05 vs. vehicle-treated controls.

**Figure 3 jcm-11-07162-f003:**
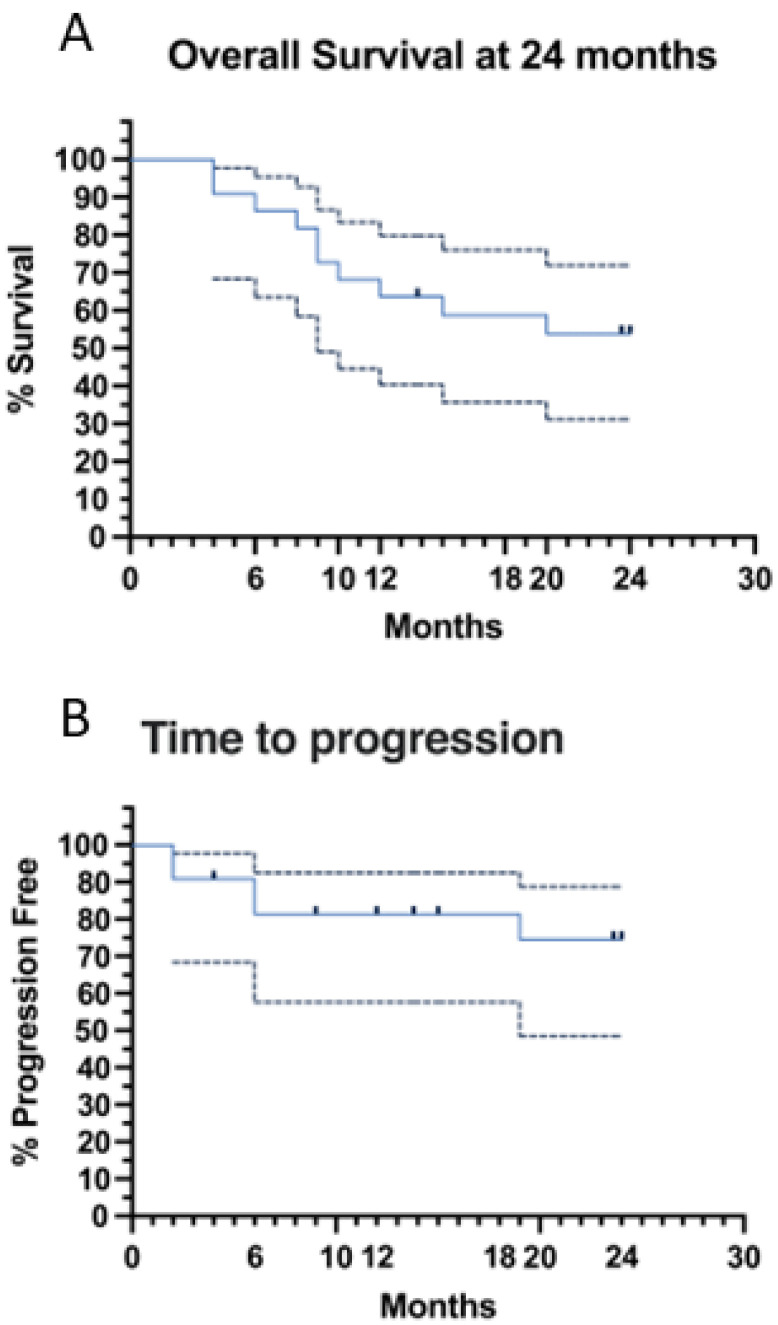
(**A**) Overall Survival (OS) estimated at 24 months is 54%; (**B**) Time to Progression (TTP) estimated at 24 months is 74%.

**Table 1 jcm-11-07162-t001:** Dose reduction index (DRI) values of Vinorelbine and Etoposide for fraction affected (Fa) of OCI-LY3 cell proliferation ranging from 0.60 to 0.99. The concomitant combination treatment of vinorelbine and etoposide was administered at the molar ratio of 1:100 for 144 h. The DRI defines the degree of dose reduction that is possible in combination for a given degree of effect as compared with the concentration of each drug alone.

Dose Reduction Index		
Fraction Affected	Vinorelbine	Etoposide
0.60	4.795	1.232
0.65	5.304	1.903
0.70	5.908	3.030
0.75	6.653	5.056
0.80	7.621	9.087
0.85	8.984	18.478
0.90	11.177	47.431
0.95	15.907	217.423
0.99	34.687	6281.652

**Table 2 jcm-11-07162-t002:** Clinical features and Response in 22 Elderly/Frail patients with DLBCL, treated upfront with R-DEVEC.

Factors	B-cell Lymphoma (n = 22) n (%)
Median Age (range)	84.5 (77–93)
Male	14 (64)
Stage III-IV	19 (86)
Ecog PS 1	5 (22)
PS 2	13 (59)
PS 3	4 (18)
Hemoglobin <12 g/dL	11 (50)
Albumin <3.5 g/dL	11 (50)
Bulky ≥7.5cm [21]	9 (41)
IPI 1–2	5 (22)
IPI 3	5 (22)
IPI 4–5	12 (54.5)
^a^ CGA	Unfit = 1(4)
	Frail = 15 (68)
	Super-Frail = 6 (27)
^b^ EPI	Int = 5 (22)
	High = 17 (78)
Histologic subtypes	DLBCL NOS = 2(9)
	Non-GC-type = 9 (41)
	GCB-type = 7 (32)
	Transformed-NHL = 3 (14)
	CHL/DLBCL = 1 (4)
Interim Response	ORR = 17 (77)
	PR = 10 (45)
	CR/CRu = 7 (32)
	NR = 3 (14)
	NE = 2 (9)
Final Response	ORR = 14 (64)
	PR = 0
	CR = 14 (64)
	NR = 4 (18)
	NE = 4 (18)

IPI 1–2; 3 and 4–5 international prognostic index score of 1–2; 3 and 4–5. ^a^ CGA: comprehensive geriatric assessment as defined by Merli and co-workers. ^b^ EPI: Elderly-IPI, from the Elderly Project of the Fondazione Italiana Linfomi by Merli and co-workers. DLBCL-NOS: diffuse large B-cell lymphoma without other specifications, Non-GC type: non germinal center type; GC-type: germinal center type on the basis of the Hans ‘algorithm [19]. CHL/DLBCL= histology with feature of both Hodgkin and non-Hodgkin lymphoma, ORR = Overall response, PR = partial response; CRu = Complete remission undetermined; NR = non-respondent; NE = not evaluable.

**Table 3 jcm-11-07162-t003:** Univariate analysis for OS, EFS and PFS of several risk factors.

Factors	EFS and OS			TTP		
	Odds Ratio	95% CI	*p* Value	Odds Ratio	95% CI	*p* Value
Hb <12 g/dL	1.167	0.2475–5.276	1	1.125	0.2120–5.919	1
Bulky >7.5 cm (n = 20) *	6.222	0.8219–31.78	**0.0861**	10	1.136–127.1	**0.0635**
DEVEC-light	1.429	0.2653–7.492	1	0.48	0.03458–4.368	1
No maintenance cycles following CR (n = 14)	2	0.08546–42.12	1	0	0.000–3.871	0.5055
EPI high	+∞	1.207–+∞	**0.0457**	+∞	0.4506–+∞	0.2725
Low albumin	0.6857	0.1427–3.442	1	0.9167	0.1423–6.078	1
Male sex	0.3673	0.07937–2.258	0.4015	0.1778	0.01380–1.408	0.179
IPI 3–5	+∞	1.207–+∞	**0.0457**	+∞	0.4506–+∞	0.2725
Not CR as Intermediate Response (n = 21)	6	0.7510–77.24	0.1736	+∞	1.015–+∞	0.0609
Not CR as Final Response (n = 18)	+∞	2.397–+∞	**0.0049**	+∞	2.397–+∞	**0.0049**
Super-frail	0.9643	0.1924–5.171	1	0.3667	0.02713–3.089	0.6214
Age > = 80	3	0.3652–42.70	0.594	0.2857	0.03816–2.418	0.2919
PS = 3	3.667	0.5689–22.33	0.3413	4.667	0.7269–26.94	0.2786

CI: Confidence Interval; CR: Complete Response; EFS: Event Free Survival; EPI: Elderly Prognostic Index; Hb: Hemoglobin; IPI: International Prognostic Index; OS: Overall Survival; TTP: Time to Progression; PS: Performance Status. *: in parenthesis is reported the number of patients for whom the factor was available.

## Data Availability

The data are available at the Department of Clinical and Experimental Medicine, University of Pisa (Prof. G. Bocci) and at the Hematology Unit, Fondazione Policlinico Tor Vergata (Dr. MC Cox).

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
