# Peer review of "Remarkable Remission Rate and Long-Term Efficacy of Upfront Metronomic Chemotherapy in Elderly and Frail Patients, with Diffuse Large B-Cell Lymphoma"

_jcm, 2022, doi:10.3390/jcm11237162_

Round 1
Reviewer 1 Report
The authors used all-oral metronomic R-DEVEC as a first line therapy in 23 elderly/frail DLBCL patients. In vitro experiments on cell line OCI-Ly3 showed synergism of vinorelbine and etoposide. The results of the clinical trial were favorable in this patient group. Important research, well performed, a few questions remain.
1. Why was OCI-Ly3 selected? There is quite a few DLBCL cell lines that might respond in different ways.
2. Figure 1 seems to say in 3 ways the same thing, maybe choose the easiest to assess? In addition a graph with the responses to the treatment showing the tryphan blue counts fort he single and combination treatment would be a very informative addition.
3. Table 1 needs more explanation, it is not clear where the 15 fold and 217 fold dose reductionare coming from.
4. In figure 2: how was the % apoptosis calculated, the vehicle is really high, realistically you should not show more than 100% apoptosis?
5. In the methods the Hans Algorithme is mentioned but there seems tob e no results of that in the tables or tekst.
Reviewer 2 Report
The authors described the results of a metronomic chemotherapy in Elderly and Frail Patients, with Diffuse Large B-Cell Lymphoma (DLBCL). The manuscript is intriguing. However, there are several issues some of which might be inappropriate for publication in Journal of Clinical Medicine.
1) Why the number of patients was less than that of the DLBCL patients in the previous report (Ref 17) by the authors?
2) Why progression free survival was better than overall survival in this study?
3) In Abstract) “Herein, we report results of the all-oral metronomic R-DEVEC [rituximab, deltacortene® , vinorelbine (VNR), etoposide (ETO), cyclophosphamide]”
Rituximab is not an oral agent.
Reviewer 3 Report
REVIEW OF THE ARTICLE:
REMARKABLE REMISSION RATE AND LONG-TERM EFFICACY OF UP-FRONT METRONOMIC CHEMOTHERAPY IN ELDERLY AND FRAIL PATIENTS, WITH DIFFUSE LARGE B-CELL LYMPHOMA.
Authors Bocci G et al.
This paper works with interesting topic of metronomic chemotherapy and focuses on the frail population of very old patients, who represent increasing subgroup of patients with problematic management. The clinical experiences are combined with in vitro experiments, which can give complex evidence of the drug efficacy.
I have some remarks that should be unclear or incomplete:
Abstract:
Term „chemo-free-like schedule“ – I recommend to omit this term, because it is misleading; chemo-free regimens are consisted exclusively from the targeted drugs (eg. lenalidomide + tafasitamab, ibrutinib+ lenalidomide etc). This „low-dose“ (or low-intensity) regimens (DEVEC, DEVEC light) are based always on chemo despite very low doses. I think, that term „chemo-free“ is widely considered as „strictly without any chemo-component“. I would advise the authors to use term „low-dose“ chemotherapy, which perfectly fits for this situation.
Introduction:
Page 2, second paragraph – the same problem with the definition of „chemo-free“, please correct.
Page 2- the use of „Deltacortene“ which is in the principle generic prednisolone. The use of trade marks should be omitted at all. Please, add at least the generic name to Deltacortene.
Despite published composition of DEVEC, I think, that it would be very practical to describe exactly dosage and time schedule in the paper, the same for „maintenance“ and „DEVEC-light“.
Methods:
Page 3 - Paragraph „Clinical study“
In the manuscript, there is written that it is a multicentric, retrospective study, but the following description of therapy, maintenance indication etc. seems to be like prospective study. Moreover, how the patients could sign inform consent in the retrospective study (at the time of analysis, more than 65% were died)? Please, explain.
Moreover, there is not evident period of enrolment or selection, and also how the patients were selected. If study is multicentric, the number of patients (n=23) seems to be relatively low. How was the proportion of the study population to all elderly DLBCL patients? Maybe selection could be bias of study (and should be commented in the Discussion).
Please, improve the structure of the “Clinical study” part, there is among the information about therapy suddenly written about myc/bcl2/bcl6…..without any basic information about pathology processing (local reading, central….)
Clinical results:
I am not content with mixture of patients. Practically all were 80years old and older, with one patient of age 26 years. From biological and statistical point of view, it does not make a sense. I strongly recommend to omit this exceptional patient from the analysis.
Please, could the authors give a more detailed information about causes unrelated to DLBCL?
Could the authors give the additional information about duration of therapy. If I understand well, majority of patients received (or intended to receive) 6 cycles of R-DEVEC (á 28 days) + 6 cycles of maintenance (á 21 days), and what is it „post-maintenance phase“. That means approx. one year – with median follow up 24 months, therapy time makes substantial time of the live since diagnosis. Could the authors add the data and comment this fact in the context of quality of life of these elderly people?
Tables are not very nice, the text is not good structured (some lines are disconnected)
Figures: there is a lot of figures, I recommend to unify for example survival figure in to one – to be graphically better arranged.
English should be corrected by native speaker.
Round 2
Reviewer 3 Report
Thank You, the authors tried to correct the text in perhaps all aspects. But, I have really problem to call this study "retrospective". According to description of all steps (including explanation to the patients) - all that is typical for prospective trial. I advise to change study to prospective multicentric (if possible) or to explain why study is retrospective.
Another think is discrepancy between the statement (line 150) - " last FU was carried out on 30th Sept 2022), and on line 185 is concluded, that last FU was on 1st Sept 2022 (if it is valid for imaging only, I recommend to omit this detail).
